# Bioenergy Potential of Crop Residues in the Senegal River Basin: A Cropland–Energy–Water-Environment Nexus Approach

**Marco Pastori** [1], **Angel Udias** [2,*], **Luigi Cattaneo** [3], **Magda Moner-Girona** [2], **Awa Niang** [4] **and Cesar Carmona-Moreno** [2]

1. External Consultant for the Joint Research Centre, GFT Italia, 21027 Ispra, Italy; marco.pastori@ext.ec.europa.eu
2. Joint Research Centre, European Commission, 21027 Ispra, Italy; magda.moner@ec.europa.eu (M.M.-G.); cesar-carmona-moreno@ec.europa.eu (C.C.-M.)
3. External Consultant for the Joint Research Centre, Piksel Italia, 21027 Ispra, Italy; luigi.cattaneo@ext.ec.europa.eu
4. Ecole Doctorale « Eau, Qualité et Usages de l'Eau » (EDEQUE), Université Cheikh Anta Diop, Dakar 10700, Senegal; awa10.fall@ucad.edu.sn
* Correspondence: angel.udias-moinelo@ec.europa.eu

**Abstract:** Access to energy services is a priority for sustainable economic development, especially in rural areas, where small- and medium-sized enterprises have many difficulties in accessing reliable and affordable electricity. Western African countries are highly dependent on biomass resources; therefore, understanding the potential of bioenergy from crop residues is crucial to designing effective land-management practices. The assessment of the capability to use crop residues for electricity production is particularly important in those regions where agriculture is the dominant productive sector and where electrification through grid extension might be challenging. The objective of this work was to guide the development of sustainable strategies for rural areas that support energy development by simultaneously favouring food self-sufficiency capacity and environmental benefits. These complex interlinkages have been jointly assessed in the Senegal river basin by an integrated optimization system using a cropland–energy–water-environment nexus approach. The use of the nexus approach, which integrates various environmental factors, is instrumental to identify optimal land-energy strategies and provide decision makers with greater knowledge of the potential multiple benefits while minimizing trade-offs of the new solutions such as those connected to farmers' needs, local energy demand, and food and land aspects. By a context-specific analysis, we estimated that, in 2016, 7 million tons of crop residues were generated, resulting in an electricity potential of 4.4 million MWh/year. Several sustainable land-energy management strategies were explored and compared with the current management strategy. Our results indicate that bioenergy production from crop residues can increase with significant variability from 5% to +50% depending on the strategy constraints considered. An example analysis of alternative irrigation in the Guinea region clearly illustrates the existing conflict between water, energy, and food: strategies optimizing bioenergy achieved increases both for energy and food production (+6%) but at the expense of increasing water demand by a factor of nine. The same water demand increase can be used to boost food production (+10%) if a modest decrease in bioenergy production is accepted (−13%).

**Keywords:** bioenergy; rural development; optimization; water; energy; food and ecosystem nexus; crop residues; sustainability; Senegal river basin

## 1. Introduction

Sub-Saharan African countries are confronted with the challenge of effectively managing natural resources to achieve higher outcomes in several sectors, such as those pointed

out in the Sustainable Development Goals [1], while ensuring sustainability and environmentally protective solutions. This challenge requires even more effort in transboundary river basins, where solutions should be balanced not only across competing sectors and scales but also taking into account the specific and eventually the different and competing development objectives of the riparian countries [2]. In regard to this, access to energy services is a priority for sustainable economic development in sub-Saharan Africa [3], especially in rural areas where limited access to energy services is accompanied by and directly linked to the other components such as food security and agricultural, water-demand, and environmental aspects. Under current trends, by 2030, about one-third of the population in sub-Saharan Africa will still lack access to electricity [4]. Although this figure represents an improvement vis-à-vis the current situation (today, the corresponding figure is 55 percent [4]), support actions are needed to meet the United Nations' goal of achieving universal access to affordable, reliable, and modern energy services for all by 2030 [5].

Energy-related greenhouse-gas emissions in Africa represent about one-third of the continental total, which is much below the global average, reflecting the importance of hydropower in electricity generation in the continent, limited industrial development, and the large contributions to emissions from land-use change and forestry, agriculture, and waste [4]. Indeed, there is great interest in Western Africa to further develop the capacity and diffusion of alternatives to fossil-fuel energy sources, which would reduce the region's carbon footprint and its dependence on oil and fossil fuels. Additionally, alternatives to fossil-fuel energy sources would reduce the impact on natural resources (wood) and would pursue political and economic goals through promoting under-utilized and domestically available resources [6]. Western African countries are highly dependent on biomass sources (such as savannah, forest, and agriculture), and, currently, biomass is intensively used but with methods that are poorly efficient and that pose a health risk [7,8]. For instance, in 2010, in the Senegal River Basin region, about 83% of people relied on firewood and charcoal for domestic cooking and heating [9]. Climate change and climate variability will also impact biomass, exacerbating the pressure on this limited resource. In addition, the population in this region is expected to double from 2010 to 2050.

Sub-Saharan Africa is noticeably endowed with a high potential for renewable energy resources such as wind, solar, hydro, and biomass, with the biomass potential estimated to be up to 1649 TWh [10]. Indeed, the potential for solar energy in the Senegal river basin is very high [11,12]; however, biomass, which is locally produced and often close to demand, could be used in tandem with hydropower and photovoltaic (PV) systems to balance the power system affected by the variable nature of solar energy. Energy potentials from agricultural residues and agro-industrial residues could be explored for present and future energy needs. In the last years, several studies have explored the potential of using crop residues to partially satisfy energy requirements, avoiding some common biofuel issues [13–22]. Specifically, in countries and regions with a dominant agricultural sector, the advantages of valorising such resources, without impacting land use or introducing negative effects on crop availability and markets, have been pointed out [23,24].

In recent years, the water, energy, food, and ecosystems (WEFE) nexus approach has taken a central stage as an integrative approach for both management and governance across multiple sectors (e.g., agriculture, food, fishing, livestock, energy, forest protection, water quality, etc.), and its concept has rapidly expanded [25–27]. The WEFE approach clearly requires policies stimulating the appropriate and efficient use of natural resources across all sectors and regions. However, it is not evident how to simultaneously maximize the benefits for all sector and regions. Therefore, advising local managers and stakeholders on how to optimize their decisions for each combination of crops and regions becomes a key element.

Optimization methods have been applied in the field of agriculture management practices [28–30] and in the application of water–food–energy–environment nexus approaches [31,32] where the optimal allocation of interlinked resources is promoted. In the specific case of biomass management [33], optimization techniques have been especially

used for supply chain optimization [34,35], with a major focus on forest biomass [36,37], while optimization techniques applied on herbaceous sources are scare in Africa [38], being even less diffused in Africa [39].

The most frequently used optimization technique in WEFE applications is the linear programming (LP) method [40], in which the objective function and all constraints are linear combinations of continuous variables. A multi-objective optimization approach [28,32] can provide a better understanding to assess bioenergy potential together with complex interlinkages required by the WEFE nexus. A nexus multi-objective analysis based on local physical resource constraints and agricultural management of bioenergy systems helps to identify context-specific solutions and implications. Indeed, a separate specific analysis of each sector (water, energy, agriculture, food) and the regions involved (Guinea, Mali, Mauritania, Senegal) in this study would result in the identification of optimal solutions just for one sector or for a country. The specific analyses for each sector are not compatible with the transboundary character of the Senegal river basin. A WEFE-nexus-integrated and inclusive transboundary management approach will address effectively the management and assessment of the multiple targets. The optimized Nexus approach requires the selection of one alternative among others, a difficult process, especially if the alternatives are non-dominated for a given set of criteria. An alternative is defined as dominant only if it is the best solution when considering all decision criteria concurrently. However, this is often not possible in a typical WEFE problem as no single alternative will be dominant due to the conflicting nature of the WEFE components. As an example, an increase in energy production and an intensification of agriculture productivity could lead to an increase in water depletion and degradation, which consequently could endanger ecological health and impact (positively or negatively) food demand satisfaction [41].

Our assessment aims to support decision making on the allocation of agricultural land for food and bioenergy production. In particular, the specific goal of this study was to integrate a bioenergy crop residue potential estimation model [42] with an optimization method [43] that assess the impact of alternative agricultural and cropland management scenarios in the Senegal river basin. The bioenergy resources are derived by agricultural residues and not by agricultural products itself, as in the case of biofuels. In this sense, this approach should be seen as a method to optimize residual materials without impacting new land use and agricultural productivity. Residue-based biofuels, however, are not automatically environmentally benign nor do they ensure the development of a sustainable energy supply.

The main questions addressed in this study include:

1.  Which bioenergy resources are available in the Senegal river basin?
2.  How should the availability of agricultural residues to satisfy the energy demand from a WEFE nexus perspective be assessed?
3.  How do strategies that maximize bioenergy production from crop residues (and vice versa) in different agricultural settings impact food production?
4.  Which are the positive and negative impacts of producing bioenergy by crop residues on other WEFE aspects? These aspects include food demand and diet satisfaction cropland allocation, water demand, and the contribution of pressure on forest and savanna environments.

## 2. The Case Study and the Baseline Scenario Definition

The Senegal river is the second longest river (1800 km) in West Africa, and its transboundary drainage basin covers about 410,000 km$^2$, over Guinea, Mali, Mauritania, and Senegal (10, 54, 26, and 15% respectively). Born in the Fouta Djallon massif in Guinea, the Senegal river travels across Guinea and Mali and, after the confluence of the Bafing, Bakoye and Falémé rivers, traces the border between Mauritania and Senegal until it meets the Atlantic ocean near Saint-Louis in Senegal (Figure 1). The journey of the river constitutes a lifeline for the 7.5 million of people of the basin (16% of the riparian countries' population) but also for the economy of the riparian countries and the region. Due to the high depen-

dency of the main livelihoods in Senegal River Basin (SRB) on water (agriculture, livestock, fisheries), around 85% of its population lives close to the river [44]. The SRB is highly vulnerable to climate variability and changes, due to the great interdependence between climate and socioeconomic activities, and it could be further challenged by the increasing pressures posed by its population dynamics on natural resources, the subsequent changes in land use and the competition among sectors and users. There is a high hydropower potential in the basin and even if currently only two plants are being exploited (one under development), the four riparian countries and the Organisation pour la Mise en Valeur du fleuve Sénégal (OMVS) have planned to increase the number of reservoirs, in order to meet the expected growing demands as well as to regulate the high inter- and intra-annual water availability of the basin [45]. In the middle valley and delta, agriculture, pastoralism, and fishing are the main activities. All of this region is poor and extremely dependent on the flood-related cropping activities in the depressions along the river for food security [46].

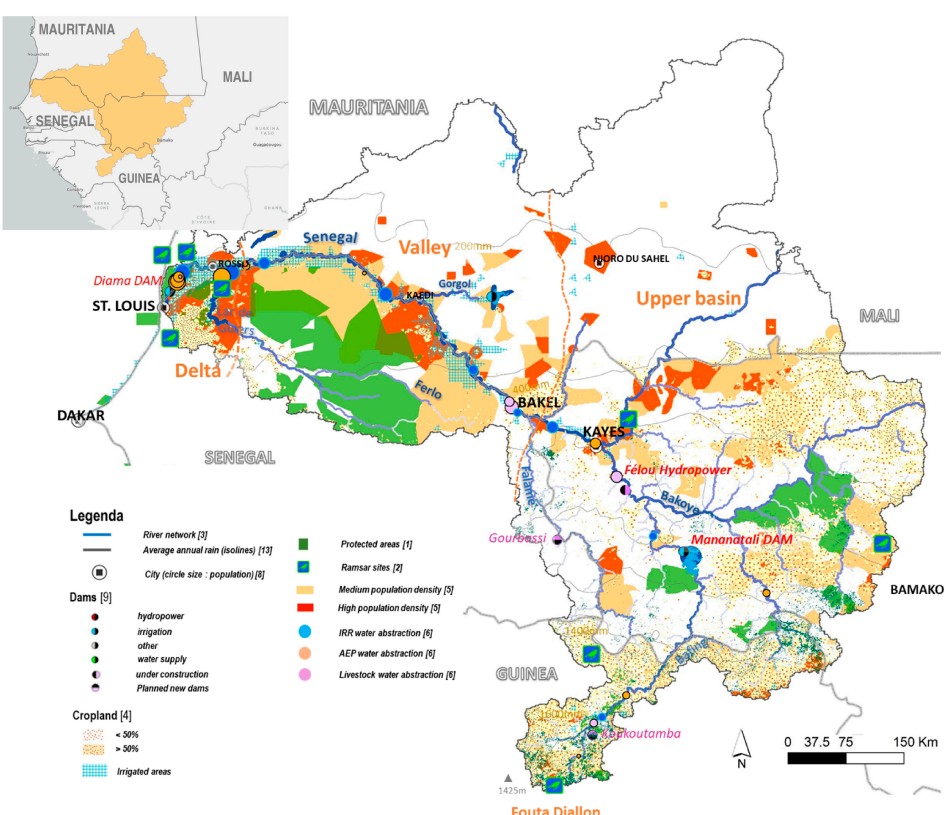

**Figure 1.** The Senegal river basin.

Cereals (like sorghum, fonio, millet, and maize) are the dominant crop types used across the SRB, accounting for about 51% of the total harvested area. Maize alone represents 8% of the total area. Other important crops are oil crops (16%), pulses (12%), rice (7%), and cotton (6%). Crops much less diffused, but anyway playing an important role in food items production, food security, and household income, are vegetables (3%) and fruits (3%). Crop productivity in the basin is quite limited, even considering that rainfed crops constitute the dominant sector. Sorghum is mainly rainfed, and its productivity (tons/ha/yr) ranges between 0.4 in Mauritania (variability within the country 0.21–0.64) to a maximum of 0.9 in Mali (0.8–1.1). Rainfed maize is the most diffused (98%), and its yield ranges between 0.58 in Mauritania (variability within the country 0.50–0.73) to a maximum of 1.62 in Mali (1.2–2.5). Another dominant crop is rice, which is mostly irrigated (76%) and which has a productivity higher for Mauritania and Senegal (average yield of about 4.5 tons/ha within the range 2.4–5.5) and lower for Guinea and Mali (average of 2.7 tons/ha within the range 2.4–3.1). Even if irrigated cropping systems are low-diffused in the basin (<1% of cropland

in Guinea and Mali, 15% in Senegal, and 25% in Mauritania), agriculture accounts for
70 percent of the total global freshwater withdrawals, making it the largest user of water.
In addition, a key contribution to food production was derived from the flood recession
agriculture, a specific cropping system based on the natural flooding of soils in order to
take advantage of soil humidity, which is currently decreasing and suffering due to the
competition and demands of large hydropower systems. Cropland expansion in the river
basin is also increasing by an average annual growth rate of 4% between 2005 and 2016
(derived by [47]).

The optimization model was set up according to the current management in the river
basin. The level of crop production, as resulting from the ongoing productivity level for
several cropping systems and minimal food demand, is reported in Figure 2.

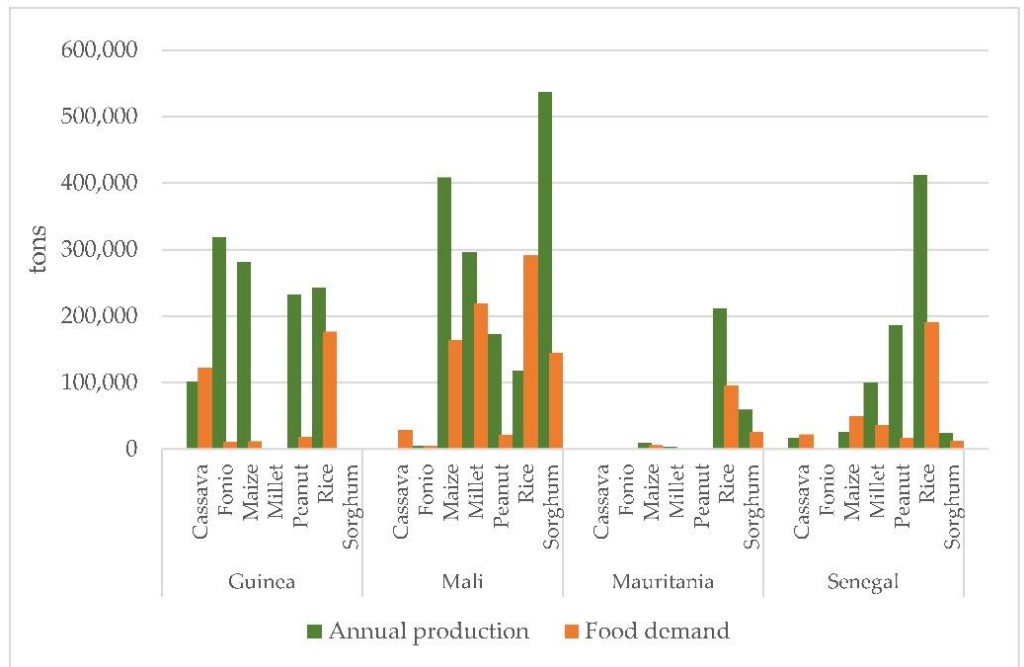

**Figure 2.** Crop production (tons) and food demand (tons) at the country level within the SRB under
current crop and land management (baseline).

The difference between food production and food demand was used to estimate an
indicator of food demand satisfaction capacity: this indicator is assessed for each subregion
and for each food item required. This indicator is important to point out which region and
which food item can be locally produced by the current farming system. The existence of
negative balance amounts does not necessarily imply a problem as the local population
can actually satisfy the food need by importing or buying products on the markets. Indeed,
if we summarize the indicator at the country level, thus assuming that all produced crops
can be distributed efficiently through the country, some of the intractable factors disappear
or are greatly reduced (Figure 3). For example, it can be seen how Mali has a very high
deficit for rice (about 0.17 M tons), which is also true for millet, maize, fonio, and cassava
(columns (a) in Figure 3); yet, most of these lacks disappear at the country level (where
the country level considered all the regions belonging to Mali but within the river basin),
while, for rice and cassava, they persist even if they may be reduced (−35% for rice and no
change for cassava). A similar tendency characterizes Guinea, Mauritania, and Senegal
where missing quantities are totally disappearing (see Mauritania) or are greatly reduced
(columns (b) in Figure 3).

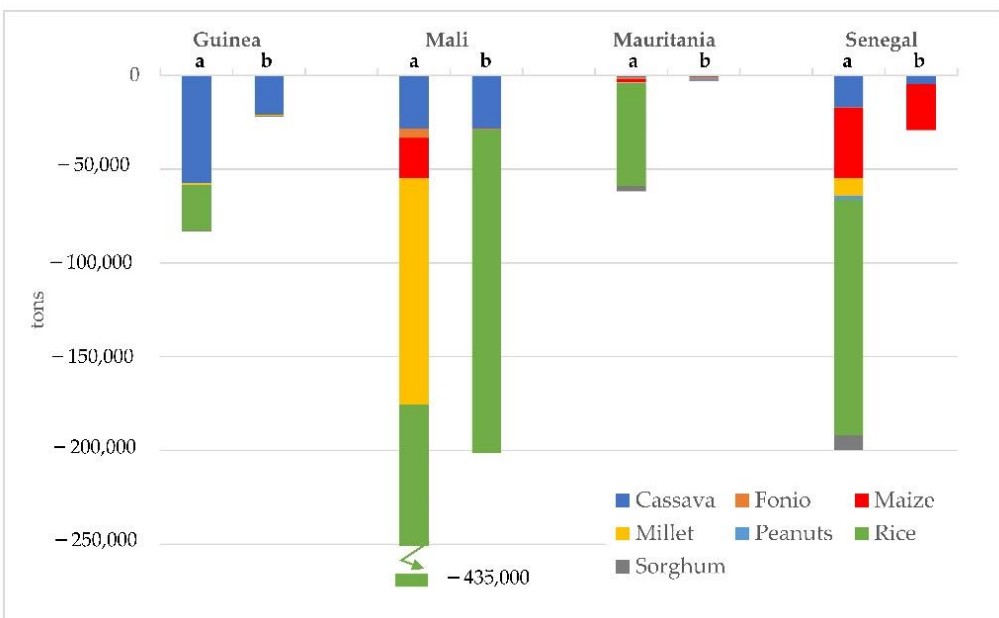

**Figure 3.** Missing food items (tons) at the country level ("self-sufficiency infectibility") as resulting by baseline crop and land management but with different repartition of crop products: (a) each subregion independent and (b) all regions together.

## 3. Materials and Methods

Under a WEFE nexus perspective, the optimization of the bioenergy production has to take into account the impact on the other WEFE sectors such as impacts on food demand satisfaction and food security, impacts on the agriculture sector (cropping requirements and livestock requirements for forage and land, water availability, and allocation), and the environment. Therefore, it is important to carefully set up and parametrize the optimization problem.

### 3.1. Design of Optimization Modelling, Objectives, and Parameters Identification

The objective of the developed optimization–modelling framework was to determine the most suitable land and water allocations by optimizing crop and bioenergy production while taking into account food self-satisfaction of local regional demand in the Senegal river basin. The expected results of the simulation process were identified according to the development priorities and specific needs as defined by local stakeholders. This analysis led to the identification of a specific set of objectives (to be maximized or minimized) and constraints to be satisfied. The purpose was to be able to identify solutions taking into account several development priorities (such as higher crop and energy production, lower water abstraction, etc.) while also identifying constraints limiting the space of variability of the optimization (constraints deriving, for example, from political issues, water planning, limits in land expansion, etc.). Figure 4 schematises the identified specific objectives and constraints of the model. These include:

- W: Minimize total irrigation water demand;
- E: Maximize total bioenergy potential from crop residues;
- E: Minimize total irrigation pumping energy demand
- F: Maximize total food production;
- E: Minimize crop land area

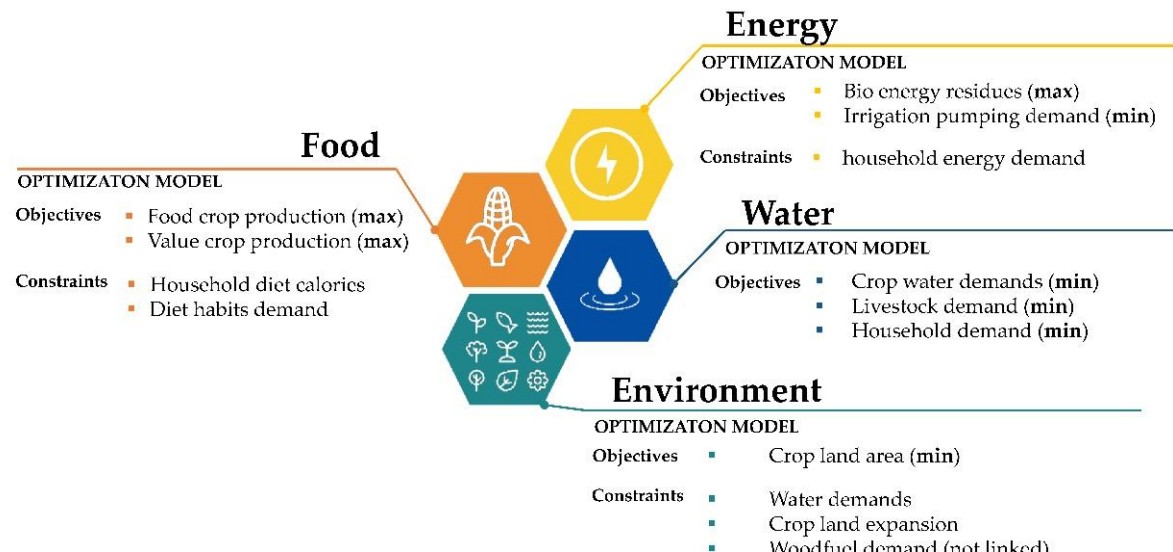

**Figure 4.** Optimization model: identification of objectives and constraints under the WEFE nexus approach. The objectives to be maximized were (F) the crop production, both for food and cash crops, and (E) the bioenergy potential from residues. While the objectives to be minimized were (W) total water demand for agriculture, (E) energy for water pumping, and (Env) crop land area.

Given this scope, a decision-oriented system was set up by using linear programming (LP) analysis tools for the assessment of crop management and land resources allocation in the river basin. The optimization model evaluates the allocation of resources (land, water, soil) in each regional area according to single and/or multiple objectives and constraints to identify optimal management strategies at local, national, and river-basin levels. Table 1 summarises the identified parameters necessary to optimise the objectives and constraints of the LP model.

### 3.1.1. The Bioenergy Potential Objective

Several crop and land management strategies are potentially possible: various crop-land allocation in specific areas of the river basin, new cropping systems, alternative irrigation and/or fertilization strategies, etc. Each strategy will derive in a direct impact on the capacity to produce more or less residues used as fuel into the power plant.

The bioenergy production directly depends on crop agricultural production and the energy potential of each type of crop residue. The annual quantity of agricultural residues generated in each administrative community in the Senegal river basin was computed based on reported cropped areas (*XRrc and XIrc* as the specific harvest area of each crop) and reported crop yields (*YieldRrc and YieldIrcc (tons/ha)* as the specific crop productivity), under current baseline management [48,49].

The objective function for maximizing bioenergy production is formulated (Equation (1)) as follows:

$$maximize \sum_r \sum_c (XR_{rc} * YieldR_{rc} + XI_{rc} * YieldI_{rc}) * EnPot_c \qquad (1)$$

where:

    *r:* specific region
    *c:* specific crop
    *XR$_{rc}$ (ha):* rainfed agriculture area in region *r* for crop *c*
    *XI$_{rc}$ (ha):* irrigated agriculture area in region *r* for crop *c*
    *YieldR$_{rc}$ (ton/ha):* yield for rain-fed crop
    *YieldI$_{rc}$ (ton/ha):* yield for irrigated crop

**Table 1.** Decision variables and parameters selected by agriculture area per region and per crop for the linear programming (LP) analysis tools.

| Subscripts | |
|---|---|
| $r$ | Region |
| $c$ | Crop |
| **Decision variables** | |
| $XR_{rc}$ | Rainfed agriculture area in region $r$ for crop $c$ |
| $XI_{rc}$ | Irrigated agriculture area in region $r$ for crop $c$ |
| **Parameters** | |
| $YieldR_{rc}$ | Specific rainfed crop productivity in region $r$ for crop $c$ (tons/ha) |
| $YieldI_{rc}$ | Specific irrigated crop productivity in region $r$ for crop $c$ (tons/ha) |
| $Calories_{rc}$ | Specific food calories content for crop $c$ in region $r$ (kcal/kg) |
| $EnPot_c$ | Energy potential for crop $c$ (MWh/year) |
| $RPR_C$ | Factor for residue production (crop-specific residue to product ratio (g/g)) |
| $RECF_C$ | Factor for residues availability (for the collection for energy production) |
| $LHV_C$ | Lower heating value (crop residue heating value (MJ/kg)) |
| $EPE$ | Efficiency conversion factor (depending on energy technology conversion) |
| $WatReqbyHa_{rc}$ | Average irrigation water requirement specific for crop $c$ in region $r$ (m$^3$/ha/cropping season) |
| $THead_r$ | Dynamic head (depth of lifting) irrigation water for region $r$ (m) |
| $PumpEff_r$ | Specific pump efficiency expressed as the energy required to lift 1 ML (mega litre) of water for 1 m of head. In this study, it was 5.9 (kwh/ML/m) |
| $AreaAvail_r$ | Total cropland area available in region $r$ (ha) |
| $WaterAvail_r$ | Water available for irrigation in region $r$ (m$^3$) |
| $MinProd_{rc}$ | Minimum production demand for crop $c$ in region $r$ (tons) |
| $MaxAreaVar_r$ | Maximum cropland area increase/decrease (% of total cropland) |
| $Pop_r$ | Population number in region $r$ (ref. year 2018) |
| $FSQ_{rc}$ | Specific food crop supply quantity per capita for crop $c$ in region $r$ (kg/capita/year) |

The technically achievable bioenergy potential (Equation (2)) was dynamically estimated based on conversion factors taking into account: (i) specific crop characteristics (linked with biomass/residues productivity and energy potential) and (ii) different types of residues derived by each cropping system and technical and sustainable usage of residues.

$$EnPot_c = RPR_c * RECF_c * LHV_c * EPE \qquad (2)$$

where:

$EnPot_c$ *[MWh/yr]*: Technical Energy Potential for crop $c$
$RPR_c$: Residue Factor Production for crop $c$
$RECF_c$: Residue Factor Collection for crop $c$
$LHV_c$ [MJ/kg]: crop residue heating value
$EPE$: Efficiency conversion factor

The relevant crop residues for bioenergy production are the *straw, stalk, husks*, and *trunks* and sometimes their *peels* after harvesting and/or processing. The theoretical potential of crop residues was estimated using specific coefficients (Table 2). Following the methodology from Kemausuor [41], a recoverability fraction (RECF) was included to reduce the effective quantity of crop residues actually available for the energy production. This factor reflects on the residues' collection availability for energy production and depends

on the type of residue. In the Senegal river basin, the RECF ranged between 0.5 and 0.7 (meaning that 50 to 70% of the residues are technically available for the energy process). This coefficient is also a way to take into account a minimum sustainable quantity of residues to remain on the soil to maintain soil fertility in the long term.

**Table 2.** Summary of average residue production and energy potentials for several crops residues as derived by literature. Sources: [39,41,42,50,51]. RPR: residue to product ratio, LHV: lower heating value as indicator of potential energy production, and RECF: recovery fraction.

| Crop | Residue Type | RPR g/g | LHV MJ/kg | RECF - |
|---|---|---|---|---|
| Mais | stalks | 2.67 | 3.50 | 0.5 |
|  | cobs | 1.00 | 4.75 | 0.7 |
| Rice | husk | 0.28 | 4.45 | 0.7 |
|  | straw | 2.19 | 3.70 | 0.5 |
| Millet | straw | 1.55 | 4.15 | 0.5 |
| Sorghum | straw | 4.15 | 4.05 | 0.5 |
| Cassava | stalks | 0.11 | 4.30 | 0.5 |
| Peanuts | shell | 2.58 | 4.90 | 0.7 |
|  | straw | 0.84 | 4.60 | 0.5 |
| Fonio | straw | 1.55 | 4.15 | 0.5 |

The values gathered in Table 2 were estimated and based on the average parametrization of residues (as derived by literature studies); however, they could vary from region to region depending on specific annual variability as they are influenced by climate and by the level of intensification of crop management (fertilization and irrigation).

3.1.2. The Food Calories Potential Production Objective

The total dietary energy (calories) objective is the result of crop productivity and their specific calorie content (Table 3), as Equation (3) formulates.

$$maximize \sum_{r}\sum_{c}(XR_{rc}*YieldR_{rc} + XI_{rc}*YieldI_{rc})*Calories_{rc} \tag{3}$$

where:

　　$r$: specific region
　　$c$: specific crop
　　$XR_{rc}$ (ha): rainfed agriculture area in region $r$ for crop $c$
　　$XI_{rc}$ (ha): irrigated agriculture area in region $r$ for crop $c$
　　$YieldR_{rc}$ (ton/ha): yield for rain-fed crop
　　$Calories_{rc}$ (kcal/kg): specific food calorie content for crop $c$ in region $r$

**Table 3.** Calories conversion used for food calories objective estimation (calories per kg of crop product, as derived by reference values at the national level provided by FAO statistics [52]).

| Crop Name | Code | Guinea | Mali | Mauritania | Senegal |
|---|---|---|---|---|---|
| Rice | RICE | 2519 | 2560 | 2908 | 2405 |
| Maize | MAIS | 3161 | 3174 | 3164 | 2997 |
| Sorghum | SGHY | 2738 | 2911 | 3081 | 2931 |
| Millet | PMIL | 2944 | 2974 | 2852 | 2451 |
| Cassava | CASS | 1086 | 1159 | 1112 | 1092 |
| Peanuts | PNUT | 3985 | 2511 | 4563 | 4169 |
| Fonio | FONI | 2301 | 3122 | 3532 | 3650 |

Food calorie requirements vary between the different countries depending on the food habits and the current diets, as defined by FAO data [52].

Equations (1) and (3) depend on the areas devoted to each crop in each region (distinguishing between irrigated and rainfed) and the productivity of each crop in each region under irrigated or non-irrigated conditions. The last factor in Equation (1) is the energy potential, and in Equation (3) it accounts for the food calories.

### 3.1.3. Total Irrigation Water and Pumping Energy Requirements Objectives

Under the WEFE nexus approach, water is a key component as several sectors are actually competing with irrigation for water resources (livestock, hydropower energy, recession agriculture, navigation, urban, etc.).

Two different objectives related to water consumption for bioenergy production were considered:

- Minimize the crop water demand (Equation (4))

$$Minimize \sum_{c} \sum_{r} (\ XI_{rc} * WatReqbyHa_{rc})$$
(4)

- Minimize the total energy required for pumping the water (Equation (5))

$$Minimize \sum_{c} \sum_{r} \left(\ XI_{rc} * WatReqbyHa_{rc} * PumpEff_r * THead_r * 10^6 \right)$$
(5)

where:

> *r:* specific region
> *c:* specific crop
> $XI_{rc}$ *(ha)*: irrigated agriculture area in region *r* for crop *c*
> $WatReqbyHa_{rc}$ *(m³/ha/season)*: average irrigation water requirement crop *c*, region *r*
> $PumpEff_c$ *(kwh/ML/m)*: pump efficiency
> *THead (m)*: dynamic head (depth of lifting) irrigation water for region *r*

Both objectives were driven for the area devoted to irrigation and the average annual (seasonal) requirements (m³/season) for each crop in each region in the irrigated surfaces.

The total energy required to pump the irrigation area depends mainly on the amount of water pumped at each point, the head difference, and the efficiency of the pumping systems [53]. This approach allowed us to have a dynamic model for energy consumption, which can eventually be specifically calibrated and downscaled for future analysis.

### 3.2. The Constraints

Jointly with objectives, constraints play a key role in defining and binding the optimization model. In this regard the choice of constraints is key to obtaining optimal solutions not only under a mathematical perspective but especially ensuring a real transferability, sustainability, and social acceptance. Diet habits demand is an emblematic example: not constraining optimal management strategies would potentially ensure high food calories support and a high bioenergy potential; however, it will be not accepted by the local population and farmers, thus preventing any efforts for their diffusion.

The constraints considered for this analysis are (Figure 4):

- Food calorie minimum requirements and diet habits
- Cropland availability
- Irrigation water availability

### 3.2.1. Minimum Food Production Constraint

This constraint was formulated using two methods, which respond to alternative possible realities: that regional demand can only be satisfied by the global production (of

all regions together) (Equation (6)) or that it can only be satisfied by the production of the sole region itself (Equation (7)).

$$\sum_r (XR_{rc} * YieldR_{rc} + XI_{rc} * YieldI_{rc}) \geq \sum_r MinProd_{rc} \ \forall \ c \ in \ C \tag{6}$$

$$XR_{rc} * YieldR_{rc} + XI_{rc} * YieldI_{rc} \geq \ MinProd_{rc} \ \forall \ c \ in \ C, \ \forall \ r \ in \ R \tag{7}$$

The minimum production ($MinProd_{rc}$) is the same in both situations as it only depends on the population and the $FSQ_{Habrc}$, as formulated in Equation (8).

$$MinProd_c = \sum_c Pop * FSQ_{Habrc} \ \forall \ c \ in \ C \tag{8}$$

The minimum production (MinProd) required to satisfy food demand was calculated by combining the diet habits of the local population with the average dietary energy (calories) requirement as defined by FAO at the national level [52,54]. Diet habits were based on the annual per capita quantity ($FSQ_{Habrc}$, Table 1) of each food crop consumed [52]. According to FAO food security indicators, each country has a specific minimum and average dietary energy requirement set up and updated annually based on specific statistics for food consumption for different population groups.

In order to incorporate all these issues into the analysis, we applied a food security weighting factor, ranging from 20% for most crops to 50% for rice and oil crops, thus reducing (starting from the production in the field) the effective quantity of food really available for consumption.

### 3.2.2. Agricultural Area Available Constraint

This constraint is imposed by the total available agricultural area in each region. The value of $AreaAvail_r$ considered in the analyses presented in this study was equal to the agricultural area available in the current scenario.

$$\sum_c (XR_{rc} + XI_{rc}) \leq AreaAvail_r \ \forall \ r \ in \ R \tag{9}$$

This constraint ensures the analysis of alternative cropland allocation while maintaining constant the total used surface: the total area (irrigated and rainfed) was kept fixed but the distribution between irrigated and rainfed was changed for each crop in each region.

### 3.2.3. Irrigation Water Availability Constraint

This constraint is meant to take into account the total water availability for the agricultural sector (crop irrigation) in each region. It does not impose any limits on how water is distributed among the different crops within each region. This constraint is formalized as:

$$\sum_c ( \ XI_{rc} * WatReqbyHa_{rc}) \leq WaterAvail_r \ \forall \ r \ in \ R \tag{10}$$

The threshold ($WaterAvail_r$) defining the maximum water use for irrigation has been currently setup according to ongoing management strategies.

### 3.2.4. Difference with the Current Strategy Constraint

In general, we can expect that radical changes in cropland area management and allocation are not desirable; therefore, a restriction factor was integrated in the model in order to allow us to control and limit the total difference areas devoted to each crop in each region, taking as a starting reference the current distribution (baseline scenario). The following equation represents such a constraint:

$$\sum_c abs(XR_{rc} + XI_{rc} - CurrArIrr_{rc} + CurrArIrr_{rc} \ ) \leq MaxAreaVar_r \ \forall \ r \ in \ R \tag{11}$$

The value of the *MaxAreaVar_r* can be defined in each analysis, and in the analyses presented in this study it was calculated as a percentage of the actual total area in each region.

*3.3. Optimization Solvers (System Implementation)*

A system has been implemented according to the optimization model described above, which allows experts to plan biomass exploitation in a region.

The optimization models were designed using the software package GLPK [55], licensed under the GNU General Public License. Since the core routines were developed in R [56], we used the Rglpk [57] package to provide a high-level solver function based on the low-level C interface of the GLPK solver. The modules of the agronomic optimizer were coded in GNU MathProg, which is a modification of AMPL [58].

The constraint method (alfa constraint) was implemented in this assessment (see [59]) to determine the set of optimal solutions that compensate for two objectives simultaneously, and it was used to construct the Pareto curve.

## 4. Optimization Results

In this section, we present key findings related to the assessment of bioenergy (electricity) potential production resulting from crop residue valorisation and its optimization taking into account WEFE-nexus indicators associated with cropland allocation and availability, water demands and availability (as based on current abstraction), and the food self-sufficiency potential.

The optimization model, described in the methods section, was applied for each country at different spatial scales and with different levels of aggregation: for each small administrative region as an independent item, at the country level by aggregating all administrative regions belonging to the same country, and at the river basin level. In this application we chose to present aggregated results at the national level because, even if they belonged to the same river basin, the four countries have important specific different characteristics (such as the food calories constraint, land allocation, and energy requirements—see Figure 2 and Table 3) and because the proposed solutions would be more adapted and acceptable for local decision makers and stakeholders.

The bioenergy, the food calories potential, and the irrigation water use resulting under the current management strategy (BLS) in each country are shown in Table 4. The management strategies here analysed are related to (i) different cropland allocation as resulting from different distributions of different crops occupying the same land area and (ii) the allocation of irrigation water available.

**Table 4.** Energy production potential from crop residues, food calories, and water use for irrigation for the current baseline (BLS) in the study area.

| Country | Bioenergy Potential<br>GWh/yr | Food Calories Production<br>Mcal | Irrigation Water Demand<br>$Mm^3$ |
|---------|------|------|------|
| Guinea | 1100 | 3270 | 70.7 |
| Mali | 1759 | 4490 | 10.5 |
| Mauritania | 750 | 2180 | 998.0 |
| Senegal | 255 | 835 | 546.0 |

Using the optimization models developed, optimal management strategies have been found that maximize (a) bioenergy production or (b) food calories production and, under different considerations, (1) the movement or non-movement of crop production between regions and (2) the different degrees of similarity with the current land allocation.

Table 5 summarizes the different optimal strategies found, indicating in which table or figure the corresponding results are included.



**Table 5.** Summary of combination of constraints considered for the optimization analysis, where: (1) FrMv/NoMv crop products can be respectively moved or not between regions, and (2) $CrL_{100}/CrL_5$ cropland allocation is totally free or is limited to a maximum of 5% of the total crop area.

| Constraints | Short Description | Crops Movement between Regions | CropLand Allocation Limit | Reported Results |
|---|---|---|---|---|
| BSL | Current management | No | 0% | Table 4, Figure 5 |
| $FrMv.CrL_{100}$ | No restriction for movement and land allocation | Yes | 100% free | Table 6 Figure 6 |
| $NoMv.CrL_{100}$ | Free land allocation but no crop movement | No | 100% free | Table 7 |
| $FrMv.CrL_5$ | No restriction for movement but limit allocation | Yes | 5% change | Table 8 |
| $NoMv.CrL_5$ | No crop movement and limit allocation | No | 5% change | Table 9 |

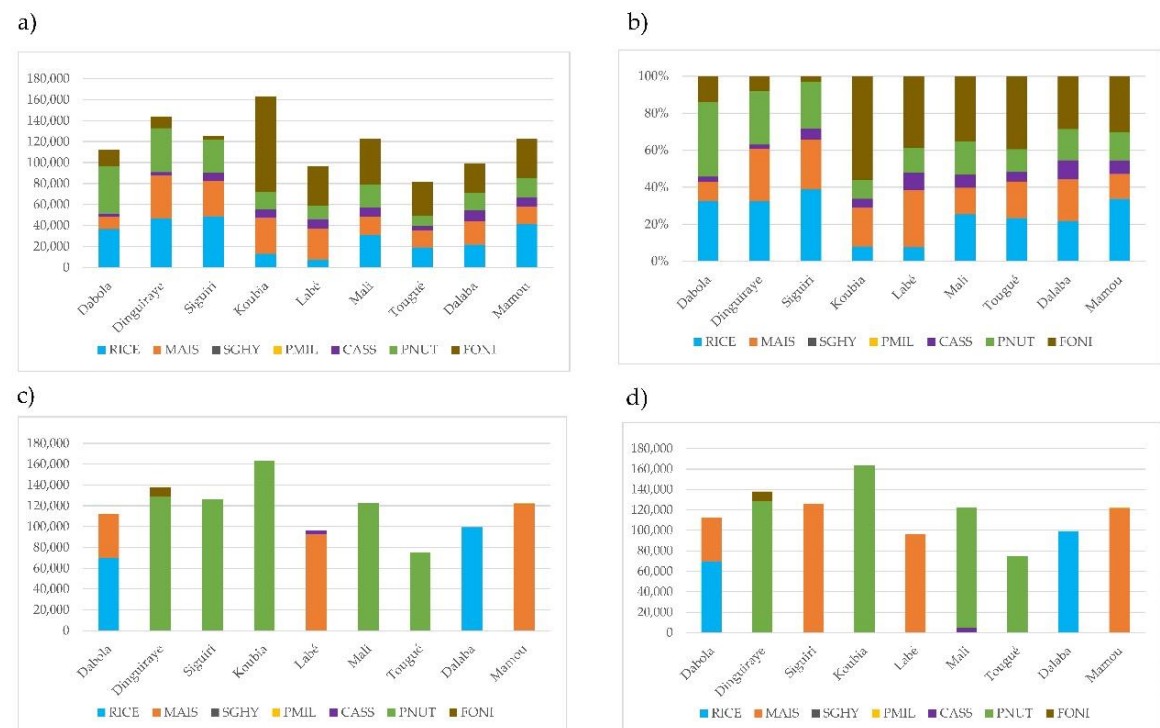

**Figure 5.** Cropland allocation for regions belonging to Guinea: (**a**,**b**) detail rainfed crop areas for the baseline as total ha and percentages within the region, while (**c**,**d**) report optimal cropland allocation according to maximization of bioenergy and maximization of food calories, under the condition of free exchange of products between regions ($FrMv.CrL_{100}$ constraints combination).

**Table 6.** Baseline (BLS)-related variations (as % changes) of bioenergy and food calories produced in each country for the strategy that maximizes energy (a) and the strategy that seeks to maximize food calorie production (b) for the $FrMv.CrL_{100}$ constraints combinations.

| Country | (a) Maximize Energy | | (b) Maximize Food Calories | |
|---|---|---|---|---|
| | Bioenergy | FoodCal | Bioenergy | FoodCal |
| Guinea | 59.6% | 36.6% | 59.2% | 37.4% |
| Mali | 57.9% | 67.1% | 39.1% | 70.4% |
| Mauritania | 60.5% | 73.7% | 54.0% | 76.8% |
| Senegal | 58.3% | 40.4% | 45.0% | 41.0% |
| River basin | 66.1% | 57.6% | 45.9% | 66.1% |

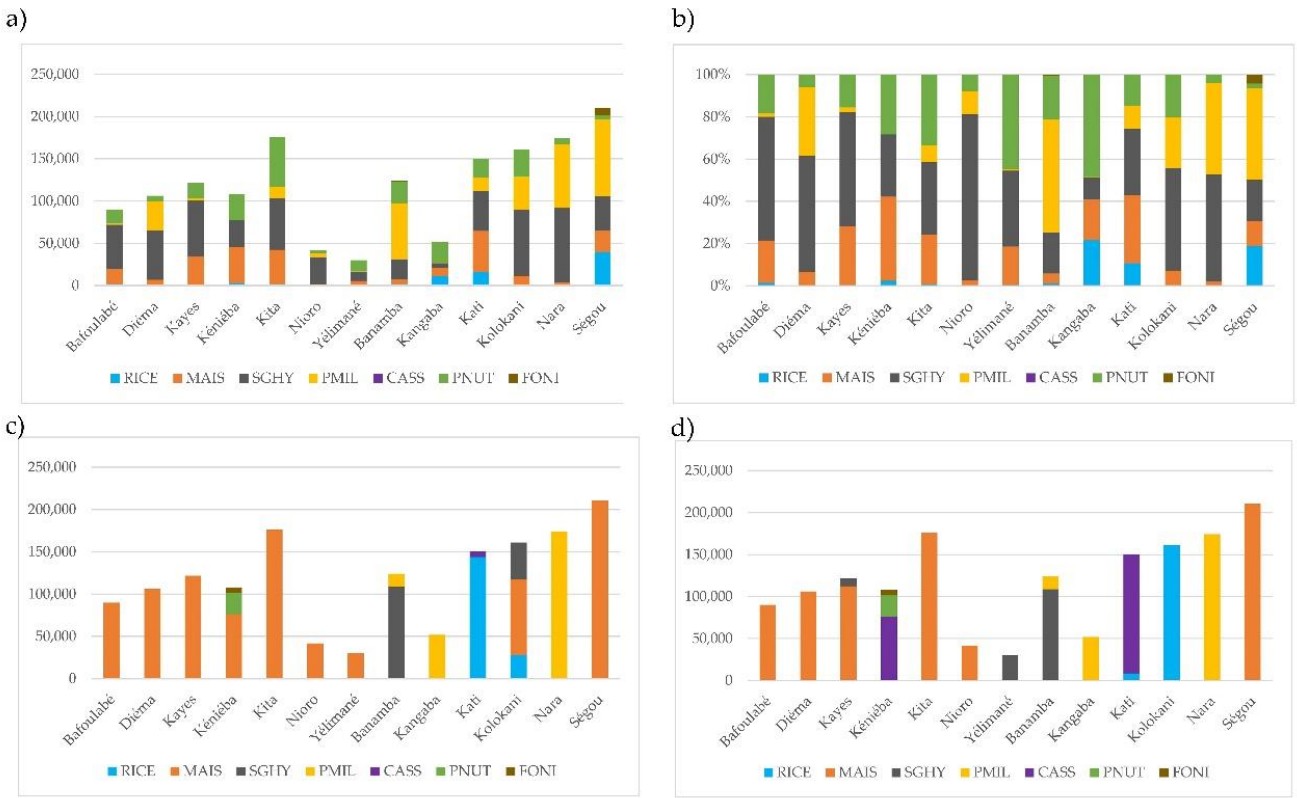

**Figure 6.** Cropland allocation for regions belonging to Mali: (**a**,**b**) detail rainfed crop areas for the baseline as total ha and percentages within the region, while (**c**,**d**) report optimal cropland allocation according to the maximization of bioenergy and the maximization of food calories, under the condition of free exchange of products between regions (NoMv.CrL$_{100}$ constraints combination).

**Table 7.** Baseline (BLS)-related variations (as % changes) of bioenergy and food calories produced in each country, for the strategy that maximizes energy (a) and the strategy that seeks to maximizes food calorie production (b) for the NoMv.CrL$_{100}$ constraints combination.

| Country | (a) Maximize Energy | | (b) Maximize Food Calories | |
|---|---|---|---|---|
| | Bioenergy | FoodCal | Bioenergy | FoodCal |
| Guinea | 46.9% | 27.0% | 30.7% | 27.4% |
| Mali | 31.1% | 40.6% | 5.0% | 46.2% |
| Mauritania | 43.7% | 57.4% | 29.1% | 58.2% |
| Senegal | 24.2% | 14.6% | 4.7% | 15.3% |
| River basin | 35.1% | 32.5% | 14.6% | 35.2% |

**Table 8.** Baseline-related variations for bioenergy and calories production for strategies maximizing (a) the energy production and (b) the food calorie production, under free movements between regions permitted (FrMv) and enabling a maximum variation rate of only 5% of total crop areas (CrL$_5$).

| Country | (a) Maximize Energy | | (b) Maximize Food Calories | |
|---|---|---|---|---|
| | Bioenergy | FoodCal | Bioenergy | FoodCal |
| Guinea | 13.8% | 8.6% | 10.8% | 9.3% |
| Mali | 6.7% | 8.9% | 3.0% | 10.7% |
| Mauritania | 6.6% | 6.3% | 3.0% | 7.5% |
| Senegal | 7.1% | 5.4% | 4.5% | 5.6% |

**Table 9.** Baseline-related variations for bioenergy and calories production for strategies maximizing (a) the energy production and (b) the food calorie production, under the constraint of satisfaction of local demand by limiting the movement (NoMv) and the sharing of crop items between regions and enabling a maximum variation rate of only 5% of total crop areas (CrL$_5$).

| Country | (a) Maximize Energy | | (b) Maximize Food Calories | |
|---|---|---|---|---|
| | **Bioenergy** | **FoodCal** | **Bioenergy** | **FoodCal** |
| Guinea | 6.5% | 3.5% | 4.4% | 4.2% |
| Mali | 2.3% | 5.9% | −0.5% | 7.3% |
| Mauritania | 3.2% | 3.4% | 1.6% | 4.1% |
| Senegal | 2.6% | 1.6% | 1.6% | 1.7% |

The most important key aspect distinguishing the different level of aggregation (independent regions vs. national and/or river basin level) is associated with the opportunity to move agricultural products between different regions. As envisaged, the strategies favouring the free transfer of crop products from one region to another are the ones that achieve the greatest increases both for bioenergy and food calorie production compared to the current situation (Tables 6 and 7).

These optimal strategies reveal a high increase, but, in reality, other constraints shall be taken into account, such as the satisfaction of food local demand, which requires a limitation to the free movement of food items and or the capability to change radically and without a social impact by cropland allocation.

When optimal strategies have to ensure the satisfaction of food demand for each region's improvements (as reported as a relative increase in Table 7), they are not as great as in the previous case (Table 6) but are nevertheless of major importance.

In both situations, two optimal strategies were analysed: (a) the one that maximize the bioenergy production using the crop residue (left Table 6 and left Table 7) and (b) the one that maximizes the crop food calories (right Table 6 and right Table 7).

Regarding the maximization of the bioenergy production, with the regional crop production sharing option (Table 6a), the optimal strategies led to a considerable improvement (about 66% at river basin level). In the regional self-satisfaction optimal strategies, with no crop items exchange allowed (Table 7a), the improvement rate was much more diversified between countries ranging between 24% (Senegal) and 47% (Guinea).

In relation to the food production (calories) maximization strategies, in the regional crop production sharing case, the increase was very large both in Mauritania and Mali (67–74%), while it was around 40% in Guinea and Senegal. For the regional self-satisfaction optimal strategies, the highest improvement rate was again in Mauritania (54%), while the lowest was in Senegal (15%).

In Guinea the differentiation of the optimization target (bioenergy vs. food production) results actually led to very similar solutions. In Figure 5, the results of cropland allocation for the two alternative optimization targets are confronted with the baseline. For both, the highest change of crop allocation occured in one single region (Koubia), where maize (a crop type important for food self-sufficiency that produces more food calories) was essentially replaced by peanuts, which are characterized by a higher energy potential. It can be seen that both optimal strategies differ significantly from the baseline, highly promoting the cultivation of peanuts over the cereal fonio. Additionally, in the baseline, rice and maize were widely produced in all the regions of Guinea, but the optimized strategies prioritized the production of each of them in specific regions. This is clearly the result of the possibility of moving crops across regions, which leads to focus the production of a specific crop where it produces more (according to current crop management and reported yield statistics). Although this figure does not show the difference in irrigation management, it is important to note that, in the optimal strategies, the cassava would be potentially irrigated, as a high difference of productivity was actually currently reported between the two production systems (rainfed vs. irrigated produce 3–4 times less). This points to how input highly affects selected strategies and the key importance of using

specific constraints to limit optimized solutions within a reasonable, social, and practical acceptable range. In no movements of the crop items allowed were the optimal strategies identified quite similar just for the conversion of fonio (even if some area was maintained) and cassava cropping in favour of peanuts; rice production in each region was retained, at least at a minimum level for local demand satisfaction.

A very different situation was found in Mali. Indeed, in this case, the differentiation of the optimization target (bioenergy vs. food production) actually led to very different solutions for the scenario configured by allowing the free movement of crops. In this case, as expected, the bioenergy target resulted in a much higher bioenergy potential increase (+60% vs. +39%), while, for food calories, there was not much difference, as both solutions identified had an important increase (Table 6). In the regional self-satisfaction case, which was also for Mali, bio-energy production increased by 31.1% or only 6.5% depending on whether it was the strategy that maximizes bioenergy or food production. In this situation the strategy that maximizes energy production seemed a better compromise than the one that maximizes calories, since the latter was much worse in terms of energy (31.1 → 6.5) and only slightly better in terms of food production (40.6 → 46.2). Figure 5 shows how in these optimal strategies a large part of the area dedicated to maize (a crop characterized by residues resulting in maximal energy production) is replaced by cassava and rice (because of their importance for food production).

The rate of improvement of one strategy is correlated with the degree of impracticability in the baseline scenario (Figure 4). This explains why the optimal strategies for Mali and Senegal provide the lower improvements, especially in the regional self-satisfaction case (Table 7).

In the case of Mali's regions, we observed a similar behaviour, with important differences between the current baseline and the identified optimal strategies. Indeed, in the baseline there is almost no area devoted to fonio and cassava, and this pattern also persists in the optimal solution that maximizes energy production; however, in the strategy that maximizes food calorie production, almost all regions in Mali devote some area for the self-production of the tuber. Other important changes in cropland allocation between the baseline and the optimised strategies could be observed (Figure 6) for peanuts, millet, and sorghum whose production was quite drastically reduced.

The optimal strategies shown in Figure 5c,d, and Figure 6c,d consider that it is possible to share production between different regions of the same country (without any loss or cost); however, transportation costs or other logistical issues could make this share not so economically interesting. Therefore, optimal strategies have also been analysed considering that each region must self-satisfice its demand. Figure 6 shows the area distributions in this situation in the regions of Mali. If we compare it with the equivalent plots in Figure 5, we can see that all the crops are produced in all regions.

A common result for all countries and scenarios, is that the identified optimal solutions require important changes in the distribution of crops on harvested land. This is clearly an aspect to take into consideration for the acceptance and transferability of the solutions to the agricultural sectors and above all to small farmers. For this reason, we have introduced another constraint by limiting the maximum variation permitted for each crop: thus, the area of one crop is allowed to increase and decrease in a limited range starting from current area, as designated in the baseline. Tables 8 and 9 and summarize increase and decrease percentage rate of the optimal strategies identified with a maximum variation rate of only 5%, and by respectively allowing or not the movement of crop items between regions. The new strategies, although as expected much more similar to current cropland occupation (baseline), produce appreciable improvements for both optimization objectives (energy and food production). Likewise evidenced in the previous analysis with no limits for area changes (see Tables 6 and 7), when it is possible to move crop production between regions, the increase both for energy and food crop production are much higher.

In addition, the missing food items quite important under the baseline management (specifically in Mali, Figure 5) were much reduced or totally disappearing for several regions.

In the free product movement between regions' optimal strategies, by its own conception, the regional self-satisfaction did not make sense to analyse the violation of regional self-satisfaction, since it promotes the search for optimal solutions at the country level without any concern for what happens in each region. Logically, regional intractable factors decreased when they can be shared between regions, as the solutions eliminate global intractable factors and increase (country) production by having much more flexibility to manage.

The intractable factors at the country level disappear in all the optimal strategies where no similarity with the current situation was required with the exception of 14,700 tons of rice in Mauritania, which the optimal strategy of regional self-satisfaction failed to eliminate.

For the optimal strategies with the limitation of 5% in maximum area variation, the degree of dissatisfaction was significantly reduced (even being null in Guinea and Senegal) or was increased depending on whether or not regional self-satisfaction was to be verified.

## 5. Discussion

There are ongoing and controversial discussions about the impacts the production of energy from agricultural biomass may have on land use, because of the competition between energy and food production, farm income, energy requirements, and greenhouse gas emissions [60]. Based on the methodology proposed in this work we focused on the valorisation of crop residues with no, or very limited, impact on the food system and on the requirement of land to produce food. This was ensured in our approach since there was no competition between the biomass use and the food sector because the optimal land-use strategies proposed should always satisfy the demand for food (for consumption by the local population), and only the residues were taken into account. In addition, the proposed approach leaves a quote of about 30–50% (depending on the crop type) of the residues on the soil, to avoid the potential negative effect of soil nutrient fertility depletion.

Another key aspect to be considered is the practical transferability of the proposed solutions and their acceptance. In order to ensure their acceptance, we introduced in the optimization models a constraint that limit the differences between new and current cropland patterns (see Scenarios 3 and 4). In the case that decision-makers only support strategies that are very similar to the current strategy, as shown in the results section, the required constraint still ensures the achievement of an important bioenergy production increase and the complete food demand satisfaction by local production, at least for the riparian regions of Guinea and Senegal (see Tables 8 and 9).

The capability of completely satisfying local food demand while enhancing energy production is highly dependent on the availability of crop residues. Clearly, the spatial dimension and the spatial level of aggregation of such residues severely affected the results. Indeed, when the movement of crop production between regions was not limited (assuming, for example, low transport costs, good conditions for product storage, etc.), the optimal strategies (see Tables 7 and 9) achieved large improvements both in food and energy production and ensured the satisfaction of all local production food demand at the country level. Concerning the temporal resolution, our approach was based on reference data for a given period (for example, for nutrition requirements, water availability, crop productivity, and exploitation of land) and year-by-year variability was aggregated in the output presented. Nevertheless, the tool can easily update the reference data to analyse new and changing boundary conditions (for example, by changing water availability, crop productivity, or nutrient requirements as affected by development).

However, the regional self-satisfaction capacity (limited movements at the regional level) remains a desirable objective, both for reasons of transport and storage cost, potential difficulties in crop transportation, and as a strategic method to ensure food security for farmers. Currently, the linkage between food security and agriculture is particularly strong

in all riparian countries of the SRB, and this is reinforced by the strong dependency of the population's diet on locally produced agricultural food commodities. Nevertheless, there are other aspects that could potentially affect food crop insecurity, such as post-harvest losses, limited accessibility to markets, lack of infrastructure for food transport and storage, and crop failures because of local and seasonal conditions.

An important aspect to be considered is that the envisaged increase in efficiency of the biomass conversion in the studied region (the Senegal river basin and, more specifically, rural areas) requires a challenging technology transfer and appropriate logistics systems. These aspects are implicitly assumed by this study.

Financial feasibility is finally a key element, which is even more complex and which should be assessed at a local scale depending on the national economy and the incentive frameworks of each of the four countries. As an example, Arranz-Piera's study in Ghana [18] indicated that a 1000 kWe combined heat-power plant would not be economically viable under the 2018 renewable feed-in-tariff rates. To ensure the economic viability of the power plant, it would be necessary to increase the rates by 25% or subsidise a minimum of 30% of the initial investment cost.

Another key aspect to be considered is the environmental impact of such strategies and more specifically the potential benefit linked with reduction in the usage of wood as the primary source of energy and cooking (with impacts on health due to indoor air pollution) [8]. We did not directly include this aspect in this application in order to reduce the complexity of the optimization process. Nevertheless, the demand for fuelwood (mainly firewood in rural areas and charcoal in urban areas) was already high in the region. The total use in 2018 was about 26 $Mm^3$, and it is expected to increase steadily. Under this framework, several authors highlighted the importance of encouraging the use of agricultural waste and promoting the application of biogas technology, specifically in rural areas where clean energy access is low and where environmental pressure needs urgently to be mitigated [61–63]. Just to provide a proxy of environmental pressure of wood fuel use in the Senegal river basin, an estimation of the forest land required to satisfy the energy demand for cooking was assessed. Based on this estimation, about 428 kg of wood fuel would be required per person, which means 20,000 $km^2$ of forest to satisfy the demand of all the rural population in 2016 (of which 40% was for Mali, 25% was for Senegal and Mauritania, and 10% was for Guinea). These and other aspects can be included in a multi-objective analysis and will be included in future studies, depending on data availability and stakeholder interest. Indeed, the results of this work indicate the importance of integrating several objectives and constraints into the optimization, for the effective identification of sustainable and viable management strategies.

Several alternative management scenarios are under development by riparian countries in the SRB [64]. Under these scenarios, small farmers will need to be adapted to depend even more on rainfed production due to an increased competition with other sectors for water and land resources potentially impacted by climate change, climate variability, and increasing demands. Indeed, crop residues' valorisation for the local production of energy is a way to increase household resilience to water availability stress and increased competition. Crop residues used for electricity production have the potential of improving electricity access rates, providing more capabilities for water pumping, and eventually groundwater resources, and for the use of alternative irrigation strategies (such as complementary irrigation) to make food production more stable and sustainable in the long term.

Certainly, the specific productivity of irrigated cropping systems is usually considerably higher than rainfed crops, both for the reduction in water stress and also because irrigation systems are normally associated with highly intensive crop management (such as fertilizer input, pest control, and the harvest and post-harvest processes). Nonetheless, irrigation cropping systems are not widely diffused in the SRB (about 6% of cropland is classified as irrigated [65]) and are mainly taking place downstream of Bakel in the Mauritania and Senegal riparian regions. Even with these small percentages of irrigated

cropland, water abstraction for irrigation is already the highest at the river basin level (1400 M m$^3$/yr) compared with residential, industrial, and livestock water demand (total of 204 M m$^3$/yr) [64]. In all the strategies presented in this study, the total amount of water used for irrigation was considered constant, to simplify the analysis and reduce the variables affecting the results. However, it is important to point out that both the increased water for irrigation or its distribution among the different crops would result in significant variations for energy and food production potentials.

This effect is illustrated by applying multi-objective optimization for the identification of trade-offs between two optimal solutions by considering the different nexus components (WEFE). The example results are shown in Figure 7, where the grey dots are the Pareto optimal front between food production and bioenergy production for a fixed amount of irrigation water. Starting from the optimal strategies under free movements between regions permitted in Guinea, optimal strategies are sought for scenarios in which the total amount of water applied increases.

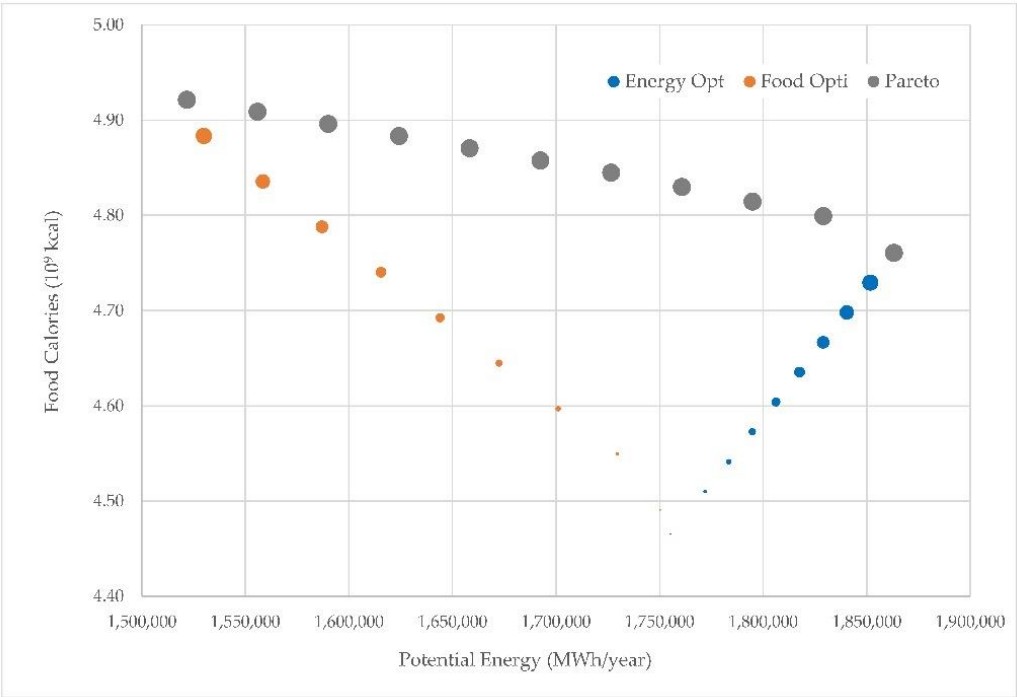

**Figure 7.** Optimal strategies identified for Guinea's river basin regions under alternative irrigation water scenarios (variable total water use for irrigation). The blue dots are the strategies that maximize the bioenergy production, and the orange dots are the strategies that maximizes the food calorie production. The grey dots are the Pareto front optimal strategies between bioenergy and food production (multi-objective) for the same water availability. The size of the dots represents the relative irrigation water quantity applied.

The optimal strategies that maximized the bioenergy potential (blue dots) achieved increases of 6.1% and 6.4% for energy and food production, respectively, when the amount of water devoted to irrigation was increased by a factor of nine (upper-right point in Figure 7). However, when the objective was maximizing food calories production, for the same increase in water, the increment in food production was 9.6%, but such a strategy implies a reduction in bioenergy production of 13% (upper-left point). This example clearly illustrates the conflicts between water, energy, and food.

## 6. Conclusions

This study developed a methodological approach by taking into consideration the context-specific land management requirements in the Senegal river basin and by pri-

oritizing options that co-deliver multiple benefits while minimizing the trade-offs of bioenergy deployment. Specifically, the methodology developed integrates optimization techniques with biophysical models in order to simulate the energy production potential from the agricultural biomass (residues), food production, and the effects of cropland management strategies.

We estimated that 7 million tons of crop residues were generated in 2016 in the Senegal river basin, resulting in an electricity potential of 4.4 million MWh/year. Several sustainable land-energy management strategies were explored and compared with the current management strategy. Our results indicate that bioenergy production from crop residues can increase from 5% to +50% depending on the strategy constraints considered without necessarily resulting in a lower production of food items and outside requiring a higher pressure on the land-use system. Under the optimal strategies that maximize the bioenergy potential, the energy and food production increased by 6%, while water use increased by a factor of nine. As a general outcome, the current baseline analysis points out that there is much space for searching for alternative management strategies to optimize local food demand specifically in Mali and Senegal, where the highest values were observed in the baseline and also partially in Guinea; Mauritania seemed to have less of an issue for this indicator, even if this does not imply that a more efficient use of cropland can be reached.

The adoption of optimization techniques was particularly useful for the development of a decision support system and assessment as they provide managers with multiple efficient strategies, which take into account different aspects, such as the degree of regional self-satisfaction of the demand or the greater or lesser continuity/similarity with the management that has been carried out in recent years; they also seek to maximize the objective of energy or food (calorie) production. This case study in the Senegal river basin is particularly complex because of the variability of climate, culture, and food habits, resulting by the transboundary character of the basin and implying the harmonization of strategies of four different countries. The comparative analysis of the optimal strategies allowed the quick identification of alternative management strategies, implying possible small modifications but with high benefits (which maximizes the probability of being easily accepted by stakeholders and, above all, by farming systems). For instance, in the Senegal river basin case, the application of this integrated system showed how it is possible to significantly increase both agricultural and energy production with limited trade-offs between them (see, for example, scenario NoMv.CrL$_5$ and the results reported in Table 9). The comparative analysis allowed us to clearly identify which was the maximum achievable improvement or target and to define effective intermediate strategies, which are able to provide important energy and food improvements while minimizing land management change. For example, in the case of Mali, the increase of about 3–5% in the maize and rice area was enough to ensure both an increase in food and energy production (+2–6%). It is also very useful for detecting the main weaknesses and vulnerabilities of current management for instance. All this allows managers or decision makers to act with greater knowledge of the situation and, above all, to have a wide range of efficient alternatives with which to make a final decision that adequately meets the interests of several sectors.

**Author Contributions:** Conceptualization, M.P., A.U. and C.C.-M.; methodology, M.P. and A.U.; software, A.U., L.C. and M.P.; validation, M.P., A.U., C.C.-M., M.M.-G. and A.N.F.; writing—original draft preparation, review, and editing, M.P. and A.U.; review, C.C.-M., M.M.-G., L.C. and A.N.F.; supervision, C.C.-M. All authors have read and agreed to the published version of the manuscript.

**Funding:** This work is a part of the WEFE Senegal project "Project appui a la gestion des ressources en eau et du Nexus Eau-Energie-Agriculture dans le basin du fleuve Senegal", which is funded by the European Union and the Italian Agency for Development Cooperation.

**Institutional Review Board Statement:** Not applicable.

**Informed Consent Statement:** Not applicable.

**Data Availability Statement:** Restrictions apply to the availability of data for cropland allocation and crop productivity. Data was obtained from OMVS and the four Countries involved in the WEFE Senegal project. DATA are available from the authors with the permission of OMVS and Countries.

**Acknowledgments:** A specific thanks to the OMVS institution for the technical and data collection support.

**Conflicts of Interest:** The authors declare no conflict of interest. The funders had no role in the design of the study; in the collection, analyses, or interpretation of data; in the writing of the manuscript; or in the decision to publish the results.

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
