# Peer review of "Bioenergy Potential of Crop Residues in the Senegal River Basin: A Cropland–Energy–Water-Environment Nexus Approach"

_sustainability, doi:10.3390/su131911065_

Round 1
Reviewer 1 Report
Dear Authors, the paper represents an interesting approach on the goven topic.
The abstract provides a good representation of the paper, however highlighting results in this section would make the research more tentative.
The intoduction is significant and comprehensive, it is carefully written. The given questions are quite demanding and since the topic is not critically reviewed in the introduction, they are difficult to address.
Materials and Methods are weel described and reproducible, however please consider to add a paragraph on DOE, which would improve the reading of the text.
The data given in tables should be either cited or SD should be provided, as it seems control is missing and the information regarding replicates of the experiment are not attached.
Optimisation model setup is described properly and significant parameters and equations are defined and explained accurate.
Statistical analysis of the data is missing, even though the disscussion is caried properly and the conclusions are supported by the results.
The given questions were addressed.
Author Response
Dear Authors, the paper represents an interesting approach on the goven topic.
The abstract provides a good representation of the paper, however highlighting results in this section would make the research more tentative.
Thanks for this suggestion. Indeed, we agree that highlighting the results in the abstract section would be important for research presentation. Given also the fact that we would like to keep the abstract concise we have tried to summarize only some key results, by pointing out the total crop residues model estimation and its expected energy potential. We have now revised the Abstract and added some more results to better clarify this point as suggested (see revised manuscript specifically from line 29-38)
The intoduction is significant and comprehensive, it is carefully written. The given questions are quite demanding and since the topic is not critically reviewed in the introduction, they are difficult to address.
Thanks for this comment on the Introduction. Since the comments highlight some difficulties, we have revised the introduction, also improving the language and reducing the number of questions highlighted at the end of the text. See revised introduction manuscript specifically from lines 89-146
Materials and Methods are weel described and reproducible, however please consider to add a paragraph on DOE, which would improve the reading of the text.
Thanks for this positive comment and suggestion. In order to follow the reviewer suggestion to add a paragraph on DOE, we have slightly modified the section 2.1, reviewing the title of the paragraph and rewriting and reorganizing the text trying to better adapt the proposed schema and to highlight first the Desired objectives of the optimization developed tool, secondly to identify the expected results of such analysis and finally to present how we have parametrized such problem to take in consideration stakeholder expected results, objectives and also constraints limiting the possibility to change variables variation range
(see revised manuscript specifically from line 230-263)
The data given in tables should be either cited or SD should be provided, as it seems control is missing and the information regarding replicates of the experiment are not attached.
Thanks for highlighting this point. We have modified the caption of the tables (Table 2 and Table 3) to better explain data source and literature references used for these data. We used reference published values for the coefficient as an ad hoc measurement of such parameters was not possible and out of the scope of the project.
Optimisation model setup is described properly and significant parameters and equations are defined and explained accurate.
Thank you
Statistical analysis of the data is missing, even though the disscussion is caried properly and the conclusions are supported by the results.
Thanks for this comment. In some preliminary versions of the paper, we included tables with a more complete statistical description of the data, including, among other things, information on the areas devoted to each crop (irrigated and non-irrigated) in each region, average parcels area, productivity levels, water consumption for irrigation. In the final version, we have decided not to include these tables, in order to focus show on the scenario results analysis, that we think is really important given the objective and focus of the application. In addition, we did not want to keep the article too long (as it already includes 9 tables and 7 figures), and this data can be also inferred from the information summarised in figure 2, within the case study paragraph.
However, in this new version, following the suggestion of the reviewer, the text description within the case study section has been enriched by including more statistical information on the areas devoted to each crop in each study region and their productivity.
Paragraph added: “Crop productivity in the basin in quite limited, even considering that rainfed is the dominant sector. Sorghum is mainly rainfed and its productivity (tons/ha/yr) ranges between 0.4 in Mauritania (variability within the country 0.21-0.64) to a maximum of 0.9 in Mali (0.8-1.1). Rainfed maize is the most diffused (98%) and its yield ranges between 0.58 in Mauritania (variability within the country 0.50-0.73) to a maximum of 1.62 in Mali (1.2-2.5). Another dominant crop is rice that is mostly irrigated (76%) and with a productivity higher for Mauritania and Senegal (average yield of about 4.5 tons/ha within the range 2.4-5-5) and lower for Guinea and Mali (average of 2.7 tons/ha within the range 2.4-3.1).”
The given questions were addressed.
Thank you

Reviewer 2 Report
Reviewer report Sustainability-1383205-peer-review-v1
The present aims to guide the development of sustainable strategies that support rural energy development by favouring both food self-sufficiency capacity and environmental benefits The manuscript addresses a research topic of increasing interest providing valuable insights for the scientific community and policy-makers as well. Moreover, the manuscript is well-written providing solid justifications in terms of novelty, filled scientific gaps and limitations. In this sense, this reviewer only has minor concerns to be addressed before this manuscript can be accepted:
- In order to improve the understanding of the reader, I recommend including the description of the case study before the presentation of the method.
- Although the parameters used in the optimisation are described in Table 1, I think it would be appropriate to include a short description of those when the authors present a new equation.
- Since some parameters are subject to changes on a yearly basis (e.g., nutritional requirements, amount of rainwater), does this optimization model have a prospective component that allows the analyst to take these changes into account? Please, discuss how this temporal aspect is/could be integrated in the assessment.
- Line 644-646: "We have not directly included in this application to limit the complexity of optimization and because of the difficulty to develop a direct link. because is quite difficult define a link Nevertheless the demand for fuelwood". Please revise this sentence.
- Although the authors describe the results obtained in detail, I think it would be suitable to include further explanations on the reasons behind the national differences in the optimisation outcomes.
Author Response
The present aims to guide the development of sustainable strategies that support rural energy development by favouring both food self-sufficiency capacity and environmental benefits The manuscript addresses a research topic of increasing interest providing valuable insights for the scientific community and policy-makers as well. Moreover, the manuscript is well-written providing solid justifications in terms of novelty, filled scientific gaps and limitations. In this sense, this reviewer only has minor concerns to be addressed before this manuscript can be accepted:
- In order to improve the understanding of the reader, I recommend including the description of the case study before the presentation of the method.
Thank you for this comment. We have moved accordingly the case study description as section 2 before the methods.
- Although the parameters used in the optimisation are described in Table 1, I think it would be appropriate to include a short description of those when the authors present a new equation.
Thank you for this comment. We have included for each equation a description of the relative parameters
- Since some parameters are subject to changes on a yearly basis (e.g., nutritional requirements, amount of rainwater), does this optimization model have a prospective component that allows the analyst to take these changes into account? Please, discuss how this temporal aspect is/could be integrated in the assessment.
Thank you for this comment. Indeed you are right in point out this issue. The optimization model allows to select optimized solution that would not be adapted for one specific year but that should be adapted on a period. For this reason the annual variability is not taken into account directly. It’s nevertheless true that user can change constraints and boundaries of the optimization model to analyse specific expected changes (for example increased demand for population, reduce water availability due to climate variability, etc.). We have added in the manuscript a paragraph on this issue.
Concerning the temporal resolution, our approach is based on reference data for a given period (for example for nutrition requirements, water availability, crop productivity and exploitation of land) and year by year variability is aggregated in the output presented. Nevertheless, the tool can easily update the reference data to analysis new and changing boundary conditions (for example by changing water availability, crop productivity, or nutrient requirements as affected by development). Lines 673-678
- Line 644-646: "We have not directly included in this application to limit the complexity of optimization and because of the difficulty to develop a direct link. because is quite difficult define a link Nevertheless the demand for fuelwood". Please revise this sentence.
Thank you. Indeed we have revised the whole paragraph to increase the reading.
- Although the authors describe the results obtained in detail, I think it would be suitable to include further explanations on the reasons behind the national differences in the optimisation outcomes.
Thank you for this comment. Indeed we have focused the analysis at national level as it was considered the most suitable for our approach and to present the results of the DSS to decision makers taking into account the specify of each country without going into details for specific regions. We have now revised the text to explain this issue see “In this application we have chosen to present aggregated results at national as even if belonging to the same river basin, the four countries have important specific different characteristics (such as the food calories constraint, land allocation, energy requirements, see Figure 2 and Table 3) and the proposed solutions would be more adapted and acceptable for local decision makers and stakeholders. Lines 477-482”
